# Phytochemicals as Chemo-Preventive Agents and Signaling Molecule Modulators: Current Role in Cancer Therapeutics and Inflammation

**DOI:** 10.3390/ijms232415765

**Published:** 2022-12-12

**Authors:** Muhammad Bilal Ahmed, Salman Ul Islam, Abdullah A. A. Alghamdi, Muhammad Kamran, Haseeb Ahsan, Young Sup Lee

**Affiliations:** 1BK21 FOUR KNU Creative BioResearch Group, School of Life Sciences, Kyungpook National University, Daegu 41566, Republic of Korea; 2Department of Pharmacy, Cecos University, Peshawar, Street 1, Sector F 5 Phase 6 Hayatabad, Peshawar 25000, Pakistan; 3Department of Biology, Faculty of Science, Albaha University, Albaha 65527, Saudi Arabia; 4School of Molecular Sciences, The University of Western Australia, M310, 35 Stirling Hwy, Perth, WA 6009, Australia; 5Department of Pharmacy, Faculty of Life and Environmental Sciences, University of Peshawar, Peshawar 25120, Pakistan

**Keywords:** phytochemicals, chemoprevention, cancer, signaling pathways, cell cycle, apoptosis, inflammation

## Abstract

Cancer is one of the deadliest non communicable diseases. Numerous anticancer medications have been developed to target the molecular pathways driving cancer. However, there has been no discernible increase in the overall survival rate in cancer patients. Therefore, innovative chemo-preventive techniques and agents are required to supplement standard cancer treatments and boost their efficacy. Fruits and vegetables should be tapped into as a source of compounds that can serve as cancer therapy. Phytochemicals play an important role as sources of new medication in cancer treatment. Some synthetic and natural chemicals are effective for cancer chemoprevention, i.e., the use of exogenous medicine to inhibit or impede tumor development. They help regulate molecular pathways linked to the development and spread of cancer. They can enhance antioxidant status, inactivating carcinogens, suppressing proliferation, inducing cell cycle arrest and death, and regulating the immune system. While focusing on four main categories of plant-based anticancer agents, i.e., epipodophyllotoxin, camptothecin derivatives, taxane diterpenoids, and vinca alkaloids and their mode of action, we review the anticancer effects of phytochemicals, like quercetin, curcumin, piperine, epigallocatechin gallate (EGCG), and gingerol. We examine the different signaling pathways associated with cancer and how inflammation as a key mechanism is linked to cancer growth.

## 1. Introduction

Globally, both industrialized and developing nations perceive cancer as a serious public health problem. Approximately 18.1 million people were diagnosed with cancer in 2018, and this number is expected to rise to 23.6 million by 2030 [1]. Despite widespread awareness of the disease, progress in finding a cure has been slow and difficult. In current practice, cancer treatment involves either surgical removal or radiation therapy to remove the bulk of the disease, followed by systemic chemotherapy for maintenance. Chemoprevention employs multiple strategies for preventing and slowing the development of cancer, and reducing cancer risk, some of which are targeting angiogenesis and altering hormones and the immune system [1,2]. Cancer carcinogenesis and metastasis prevention have been the focus of several natural and manufactured agents [3]. Anti-mitotic and anti-microtubule alkaloid agents (e.g., vinca alkaloids, podophyllotoxin (PTOX)), DNA-interactive agents (e.g., cisplatin, doxorubicin, camptothecin (CPT)), anti-tubulin agents (taxanes), hormones, and molecular targeting agents [4] are among the most common chemotherapeutic drugs. They suppress cellular microtubule assembly and cell death by modulating metabolic and signaling pathways [5]. However, chemotherapy has several drawbacks that may negatively impact a patient’s quality of life, including cancer recurrence, drug resistance, and harmful effects on nontargeted organs. There is a continued search for new potential anticancer drugs with improved effectiveness and fewer side effects to address the limitations of standard cancer treatment. Phytochemicals and derivatives contained in plants are a viable approach to enhance the efficacy of therapy in patients with cancer and reduce side effects. A variety of phytochemicals are physiologically active and have substantial natural anticancer potential. Testing natural extracts (from dry or wet plant material) for potential anticancer biological activity is the first step in developing a safe and effective phytochemical-based anticancer therapy, followed by purification of active phytochemicals and isolation of active compounds via bioassay-guided fractionation, and lastly, testing them for in vitro and in vivo effects. 

To add to their potential medicinal uses, several phytochemicals are also anti-inflammatory in nature [6,7]. Inflammation can cause swelling, redness, heat, and pain, and is the body’s main defense against microbial infections, damaged cells or tissues, and radiation [8,9]. Cell surface receptors sense danger and set off the inflammatory cascade, which results in the production of inflammatory markers and the recruitment of inflammatory cells [10]. Unlike acute inflammation, which resolves after the offending factor is eliminated, chronic inflammation persists and takes longer to resolve, because the body is unable to heal completely. Cancer, heart disease, diabetes, chronic renal disease, neurological illnesses, and autoimmune problems are all caused by persistent inflammatory reactions [11]. Pro-inflammatory cytokines, including interleukin (IL)-1, IL-6, and tumor necrosis factor-alpha (TNF-α), as well as non-cytokine mediators such as reactive oxygen species (ROS) and nitric oxide (NO), are released during the inflammatory response [7]. Therefore, it is crucial to prevent and cure many illnesses by dampening inflammatory reactions. Novel anti-inflammatory compounds with high specificity and fewer side effects are urgently required. 

Some examples of sources of such phytochemicals include grapes, which contain resveratrol and quercetin [12]. The anti-inflammatory compound curcumin is found in turmeric roots [13], while piperine is derived from black peppers [14]. Among the catechins found in green tea, EGCG is a notable one [15]. The medicinal component gingerol is found in ginger roots [16]. In this review, we summarize the molecular signaling pathways related to inflammation and explore the anti-inflammatory effects of these, and a few other phytochemicals in malignant cancers, focusing on the possible inflammation-phytochemical interactions and outlining the signaling pathways involved in chemo-preventive measures.

## 2. Mechanisms of Cancer Chemoprevention 

### 2.1. Signal Transduction Mechanisms

The intricate interactions of signaling molecules [17] are one of the several variables that lead to cancer growth. These pathways are becoming more transparent as science and research methodologies develop and are used to trace the factors that contribute to the development of cancer. With detailed knowledge of gene expression, repression, mutations, etc. [18,19], molecular oncology has uncovered crucial mechanisms of action. Numerous therapies and medications have been effective in cancer treatment. However, modifications in lifestyle and food may reduce the risk of cancer and slow the disease course [20]. There is a lot of hope that these chemo-preventive substances found in food may help to treat a wide range of diseases. Many studies have been done to uncover the mechanisms of action of naturally occurring chemo-preventive compounds, mainly from plants, for cancer prevention [21]. Phytochemicals or molecules found in plants disrupt cellular processes such as oxidative stress reduction, oncogene activation, gene silencing, cell cycle arrest, apoptosis, necrosis, and autophagy, as well as angiogenesis and metastasis. Activators of chemoprevention in all these and many other pathways have been identified [22]. Kinases, receptors, caspases, tumor-suppressor proteins, transcription factors, microRNAs, cyclins, and cyclin-dependent kinases are examples of target molecules. Oncogenes are activated and carcinogenesis is triggered by a variety of internal and external events. However, cells launch their self-defense mechanisms by modifying cell signaling pathways, including apoptosis, cell cycle arrest, and autophagy. These defensive systems, which are regulated by a plethora of molecules located in the cell matrix, cytoplasm, and nucleus, may be amplified by phytochemicals [23,24]. 

### 2.2. Key Signaling Pathways 

The two main mechanisms controlling cell growth and survival are the ERK and MAPK pathways, both of which can be targeted by phytochemicals. Plant-based chemicals have been shown to be effective in preventing cancer through a variety of mechanisms, including modulating HIF-1 production, stability, accumulation, and transactivation by affecting major regulators of the glycolytic system, such as glucose transporters, hexokinases, phosphofructokinase, pyruvate kinase, and lactate dehydrogenase, with a focus on the PI3K/Akt/mTOR and MAPK/ERK pathways as critical signaling cascades in HIF-1 activation and autophagy-apoptosis pathways [25,26]. Ursolic acid, kaempferol, resveratrol, gingerol, sulforaphane, genistein, and isothiocyanates are some of the most well-studied plant chemicals and have been shown to cause apoptosis in cancer cells through the MAPK and ERK pathways [27,28]. 

The AKT/PI3K signaling pathway plays a critical role in the regulation and development of cancer. Epidermal growth factor (EGF) levels govern a range of molecular pathways, including activation of NF-κB and phosphorylation of AKT, which in turn leads to resistance to apoptosis and unchecked cell proliferation, and further downstream, it leads to the regulation of effectors, such as FOXO (fork head box) [29,30,31]. The expression of these downstream effectors is regulated by alkaloids and phenolic compounds. Cell cycle arrest and apoptosis, inhibition of AKT/PI3K signaling, FOXO3a activation, anti-proliferation, and anti-invasion are some of the anticancer effects of resveratrol, luteolin, apigenin, flavones, sulforaphane, and curcumin, respectively [32,33,34,35,36,37]. The transcription factors p53, Bcl-2, cyclin D, and interleukin-6 (IL-6) involved in cell death, proliferation, and apoptosis, are regulated by STATs, which are phosphorylated after JAK activation and then translocated to the nucleus [38]. Phytochemicals dramatically promote cell death in several cancer types by downregulating JAK/STAT signaling and activating apoptotic cascades [39,40,41]. Aberrant Wnt/-catenin signaling has been linked to many different types of cancers [42,43], including those of the breast, lung, colon, blood, ovary, skin, and brain. The transcriptional factors that control cell proliferation, survival, and migration are activated when the Wnt protein binds to frizzled family transmembrane receptors, and β-catenin accumulates in the nucleus [44]. Activation of glycogen synthase kinase 3 (GSK3) is responsible for the effects of curcumin, resveratrol, and EGCG on β-catenin translocation and accumulation in the nucleus [43,45,46,47,48]. Similarly, p53 is responsible for anticancer activity in cells, similar to other signaling pathways. This tumor suppressor protein promotes the activation of the apoptotic cascade [49]. Resveratrol and costunolide are only two examples of phytochemicals that have been proven to increase p53, lower AKT, and cause cellular apoptosis and cell cycle arrest [50,51,52]. Crosstalk between the p53, MAPK, and JNK pathways is also a major contributor [53]. Autophagy, in which cells shut down abnormal growth and advancement, is facilitated by curcumin and ursolic acid. The AKT and mTOR pathways are downregulated in cancer cells, inducing autophagy [54]. Inflammation, angiogenesis, invasion, and metastasis are controlled by dietary phytochemicals [55], which means that they also govern anti-tumor action in tissues. Figure 1 represents the signaling pathways through which phytochemicals inhibit carcinogenesis.

### 2.3. Angiogenesis Inhibition and Tumor Dissemination/Metastasis

Angiogenesis refers to the physiological process by which new blood vessels are generated from pre-existing blood vessels. This is a critical stage in the progression of cancer, during which they invade the surrounding tissue and disseminate. As angiogenesis inhibitors, phytochemicals such as phenolics may prevent tumor cell growth and proliferation. Several in vitro and in vivo investigations have provided a solid foundation for understanding their fundamental anti-angiogenic mode of action. Ellagic acid, EGCG, genistein, and anthocyanin-rich berry extracts exhibit antiangiogenic properties by inhibiting the expression of angiogenic factors, including the vascular endothelial growth factor, vascular endothelial growth factor receptor-2, hypoxia-inducible factor 1a, platelet-derived growth factor, platelet-derived growth factor (PDGF) receptor and matrix metalloproteases. Inhibiting the phosphorylation of the EGF, VEGF, and PDGF receptors can in turn inhibit angiogenesis [56,57,58]. The anti-angiogenic factors in tumors and normal cells may react differently to several phenolics. Chelating ferrous ions and inhibiting hypoxia-inducible factor-1a-induced cancer cell proliferation are two of EGCG’s many abilities. Treatment of prostate tumor cells with EGCG in vitro decreased hypoxia-inducible factor-1a protein levels and hypoxia-inducible factor-1a-mediated transcription even in the presence of adequate oxygen [59]. To stop the angiogenesis process, the flavonoids kaempferol, quercetin, myricetin, and galanin inhibit tube development of human umbilical vein endothelial cells (HUVECs) in response to VEGF and limit the adherence of U937 cells to HUVECs [60]. 

In murine T-cell lymphoma, flavonoids (which are polyphenolic in nature) such as quercetin suppress angiogenesis by lowering VEGF-1 levels via AKT signaling [61]. Cancer cells may metastasize to distant organs through the lymphatic system. Extracellular matrix breakdown, proteolysis, cell adhesion, cell migration, angiogenesis, and invasion are all influenced by this mechanism [62]. Polyphenolic compounds in foods inhibit tumor cell invasion and metastasis via many pathways. However, the molecular mechanisms and signal transduction pathways governing these processes remain unknown [63,64].

### 2.4. Cell Death and Cell Cycle Halt

Many cancer treatments focus on triggering planned cell death known as apoptosis, which is the most prevalent form of cell death. Inhibition of carcinogenesis by activation of apoptosis has been the mechanism of action of several dietary chemo-preventive agents (resveratrol, quercetin, EGCG, curcumin, apigenin, chrysin, silymarin, and ellagic acid) [65]. These chemicals have a greater effect on cancer cells than on healthy cells [66,67]. EGCG causes sarcoma cells to undergo G2/M phase arrest, downregulates Bcl-2 and myc, and upregulates p53 and Bax expression, whereas the expression of other crucial apoptotic targets, including p21, p27, Bcl-xL, mdm2, and cyclin D1, is unaffected [68]. In a study using isogenic cell lines, the activation of caspase-3 and -9 and PARP cleavage, together with the overexpression of p21, p53, and Bax, in prostate cancer cells led to apoptosis. EGCG-induced cell cycle arrest and death largely through p53-mediated signal transduction, with additional support from p21 and Bax [69].

Theaflavin, a phenolic compound found in black tea, promotes apoptosis by increasing DNA fragmentation, caspase-3 and -8 activity, Bax expression, and Bcl-2 downregulation [70]. Remarkably, theaflavin also induces apoptosis in LNCaP cells by upregulating p53 expression, which subsequently influences the NF-κB and mitogen-activated protein kinase pathways [71]. Anthocyanidins, a type of flavonoid chemical, have been shown to inhibit colon cancer cell growth in a dose-dependent manner. DNA strand breaks and unbalanced expression of Bax and Bcl-2 are responsible for the activation of apoptosis in colon cancer cells by anthocyanins. However, some bioflavonoid substances do not suppress cell proliferation [72]. These include rutin, epicatechin, chlorogenic acid, and p-hydroxybenzoic acid. The plant pigment delphinidin, which belongs to the anthocyanidin family, stops the cell cycle in the G0/G1 phase, preventing the vascular endothelial growth factor from promoting cell migration and proliferation. During this time, p21 and p27 expression increased, whereas cyclin D1 and cyclin A levels decreased sharply [73]. Early activation of ERK 1/2, overexpression of caveolin-1, and downregulation of Ras [74] also contribute to the delphinidin-mediated antiproliferative effect. 

Moreover, several studies suggest that curcumin may inhibit the proliferation of glioma cells by influencing the RB1/CDK4/p16INK4A and TP53/MDM2/MDM4/p14ARF signaling pathways. These are the two major pathways involved in cell cycle control and are often mutated in glioblastomas [75]. Curcumin dramatically inhibits the development and proliferation of many human glioma cell lines by inhibiting cell cycle progression. Specifically, curcumin causes p53-dependent G2/M cell cycle arrest. Curcumin treatment of U251 glioma cells led to increased p53 protein levels through the stimulation of p21 (cell cycle regulator)/CDK inhibitor and tumor suppressor ING4 [76]. This increase in p53 and p21 expression was coupled with the downregulation of RB and cdc2 pathways in DBRTG glioma cells in another study [77]. According to studies carried out with U87MG cells, curcumin promotes cell cycle arrest in glioblastoma cells by downregulating cyclin D1 and upregulating p21 [78]. Surprisingly, this occurs without p53 involvement and relies instead on activating Egr-1 (a transcription factor) via the ERK and JNK/MAPK/Elk signaling pathways [79]. In contrast, when the human cell line U87MG was transfected with Egr-1 siRNA, the curcumin-induced transcription of p21 was inhibited [80]. Curcumin inhibits cell proliferation by increasing the expression of the tumor suppressor death-associated protein kinase 1 (DAPK1), according to current research. This effect was observed in U251 cells, which were arrested at the G2/M phase of the cell cycle after treatment with curcumin and led to caspase-3 activation and the downregulation of the NF-κB pathway. However, the knockdown of DAPK1 with siRNA reduced the curcumin-induced suppression of NF-κB and STAT3, limiting caspase-3-mediated apoptosis [81].

### 2.5. Current Cancer Therapy Involving Phytochemicals

Epipodophyllotoxin, camptothecin (**CPT**) derivatives, taxane diterpenoids, and vinca alkaloids are the four main groups of plant-derived anticancer agents employed in clinical practice. In addition to the aforementioned groups of phytochemicals, additional plant-derived anticancer drugs such as combretastatins, homoharringtonine (omacetaxine mepesuccinate, cephalotaxine alkaloid), and ingenol mebutate are also used (Table 1). Research has concentrated on finding ways to eliminate the consequences of the phytochemicals poor water solubility and its considerable hazardous side effects [82]. Several attempts have been made to improve water solubility and tumor selectivity, leading to the synthesis of numerous analogs and prodrugs in this area [83]. The following is a quick overview of a few phytochemicals currently used in cancer treatment. 

### 2.6. Vinca Alkaloids 

Of all the medicinal plants that have been used for treatment since ancient times, *Catharanthus roseus* is by far the best known. Among the 345 bioactive phytochemicals extracted from the plant, the high concentration of indole alkaloids (vinblastine, vincristine, and vindesine) makes it a popular herbal treatment for cancer [98]. There are many different sections of the *C. roseus* plant that are utilized in Ayurvedic folk medicine for the treatment of cancer, diabetes, and liver, renal, gastrointestinal, and cardiovascular disorders. In Jamaica, West Indies, diabetes has long been treated with a hot water extract made from the dried leaves of this plant. Similarly, in Peru, people with conditions including heart disease and cancer have taken hot water plant extracts orally as a supplemental and alternative treatment [98]. Nowadays, you may buy Velban, Oncovin, and Vinflunine, to mention a few of the brand names for the *C. roseus* indole alkaloids that have been identified and brought to the market [99]. Hodgkin’s disease, lymphosarcoma, neuroblastoma, and breast cancer are only a few of the many cancer types treated with these alkaloids.

Vinca alkaloids, such as vincristine, vinblastine, and their semi-synthetic derivatives vindesine, vinorelbine, and vinflunine, are abundant in the stems and leaves of *C. roses* [100,101]. Vinflunine and vinorelbine offer enhanced therapeutic efficacy and are derived from the alkaloids, catharanthine and vindoline, respectively [102]. Children with leukemia are the target population for vincristine administration [103]. Second-line urothelial transitional cell carcinoma has been studied in mice and humans using vinflunine and vinorelbine [101]. By attaching to tubulin, these chemicals exert anticancer action and are known as “mitotic poison” [101]. 

By altering microtubular dynamics, they can suppress cell growth, ultimately leading to apoptosis [102]. Vinblastine sulfate has been used to treat a wide variety of cancers, including Hodgkin’s disease, choriocarcinoma, neuroblastoma, lymphosarcoma, and breast and lung carcinomas. Acute juvenile leukemia, lymphocytic leukemia, Hodgkin’s disease, reticulum cell sarcoma, neuroblastoma, and Wilkins’ tumors are treated by vincristine sulfate, an oxidized version of vinblastine [102,104]. The anticancer effects of vinblastine result from the interaction of the drug with tubulin, which blocks metaphase in mitosis. Vinblastine binds to microtubular proteins in the mitotic spindle, leading to microtubule crystallization, halting mitosis, and initiating apoptosis [102,105]. Due to their mechanism of action against cancer cells, a double-sided sticking process involving interactions with β-tubulin, vinblastine and its derivatives are essentially interchangeable [106]. Vindoline, vinblastine’s chemo predecessor, is also effective against cancer and is derived from tabersonine [85]. Vincristine, like vincamine, is a naturally occurring organic compound derived from the vinca plant [107]. To prevent cells from entering mitosis, this alkaloid (vincristine) functions as an anti-microtubule agent [102,108]. These drugs function by inhibiting the polymerization of tubulin, which stops the production of new microtubules and causes the depolymerization of existing tubules. The action of vincristine also inhibits the usage of glutamic acid, thus interfering with nucleic acid and protein formation [85]. Vincristine has been licensed by the US Food and Drug Administration (FDA) for the treatment of Hodgkin’s disease and Hodgkin’s non-lymphoma, acute lymphocytic leukemia, and the lymphoid blast crisis of chronic myeloid leukemia. Vincristine has been used off-label to treat a variety of cancers, including those of the central nervous system [107,108], Ewing sarcoma, prenatal trophoblastic tumors, multiple myeloma, ovarian cancer, and small cell lung cancer. 

### 2.7. Taxanes (Paclitaxel) 

Yews, or Taxus spp. of the family Taxaceae, are evergreen non-resinous gymnosperms that mature slowly over time. The English yew (*T. baccata*), Pacific or Western yew (*T. brevifolia*), American yew (*T. canadensis*), and Japanese yew (*T. cuspidata*) are the most popular species grown in gardens. Taxanes, derived from these plants, are well recognized for their ability to inhibit cell division; however, their complicated mechanism of action also causes them to interfere with a wide variety of other physiological and cellular processes [109] (Figure 2).

Microtubule-targeting anti-mitotic drugs are significant chemotherapeutic drugs since they target tubulin and other microtubule components. Microtubules play crucial roles in cell division and other key biological activities, such as cell motility, intracellular transport, and differentiation. They arise in the centrosome during the interphase and are hollow cylinders made of 13 protofilaments with a typical diameter of approximately 22 nm. Mitosis is characterized by the drastic remodeling of microtubules, including their breakdown and the development of new structures such as mitotic asters and kinetochores. This includes a carefully orchestrated chain of events ensuring that chromosomes bind and separate properly during cell division. Microtubule-associated and microtubule-interacting proteins have a significant impact on these processes [110,111]. Microtubules made from various tubulin isotypes exhibit distinct degrees of mobility [112]. Multiple post-translational modifications of tubulin, such as phosphorylation, acetylation, methylation, polyamination, palmitoylation, ubiquitination, glycosylation, arginylation, sumoylation, succination, and O-Glc-NAcylation also contribute to its diversity and microtubule biology. While acetylation, phosphorylation, polyamination, and methylation also occur in the core areas of tubulins, there is a large variation in the sequence. Microtubule dynamics and the interaction of microtubules with motor proteins, such as kinesins and dynactin are affected by such modifications. Post-translational modification enzymes exert control over a lot of these tubulin modifications [113,114,115,116]. The microtubules of α-tubulin 4a and 4b are relatively detyrosinated, which may contribute to their high stability [117]. Alpha-tubulin 1b (α-tubulin 1b) expression has been linked to a poor prognosis in lymphoma and hepatocellular carcinoma, whereas α-tubulin 3c expression has been linked to resistance to paclitaxel in ovarian cancer [118,119]. The sequence of β3-tubulin lacks a cysteine at position 239 (Cys239) but has a unique Cys124 and a phosphorylatable serine at the C-terminus, which may explain why the presence of β3-tubulin reduces microtubule stability and enhances their dynamics compared to microtubules carrying β2- or β4s-tubulin [120,121,122]. There is growing evidence that β3-tubulin expression is a reliable predictor of cancer aggressiveness, with reports of its presence in a wide variety of malignancies [123]. Since its post-translational modification pattern and its interaction with mitogen-activated protein kinases (MAPKs) are isotype-specific, β3-tubulin is more resistant to oxidative stress than other tubulins [124]. Paclitaxel decreases binding to β-tubulin dimers but strengthens binding to β-tubulin of polymerized microtubules [125]. This dampens the microtubule dynamics. The stabilization of microtubules and significant rearrangement of the microtubule network into dense tubular bundles are both effects of paclitaxel addition (in vitro and in vivo) [126]. Cell death may occur at any point in the cell cycle, from interphase through G1 to mitosis, depending on taxane content, cell type, and cell condition [127]. Taxanes cause aberrant mitotic arrest and disruption of the mitotic spindle by activating the spindle assembly checkpoint [128]. Depending on intraline and interline differences, as well as MAPK expression, many cells die during mitosis after a protracted period of arrest, while others leave mitosis without dividing and return to the interphase [129]. These analyses show that mitosis is uncommon in human solid tumors and that taxanes and other microtubule-stabilizing drugs work in many ways to kill cancer cells, except for the prevention of mitosis. 

They do not need to halt mitosis and complete their anti-mitotic activities. Mitotic slippage is an adaptation of cells to medications such as taxanes, which stabilize microtubules [130]. As CDK1 activity is required for spindle assembly checkpoint activation, cells often divide when exposed to chemotherapeutic drugs. A gradual loss of CDK1 allows cells to bypass mitotic arrest and escape mitosis since the spindle assembly checkpoint is unable to prevent the breakdown of cyclin B, which allows cells to leave mitosis [131]. Variable effects were observed depending on the amount of taxane used. Taxane therapeutic efficacy is proportional to the number of microtubules to which they are bound. The overall concentration of taxanes inside cells greatly surpasses the concentration in medium or plasma because taxanes accumulate in cells and tumors [132,133]. ∼20 μM, of tubulin, or 4% of total protein in HeLa cells, polymerizes in response to the addition of taxanes [134]. The relative importance of phenotypic effects on site occupancy was ranked by Pineda et al. The loss of MT dynamics was the most susceptible to perturbation, with detection beginning at a site occupancy of ∼0.1 at a 1 nM paclitaxel concentration. This was followed by micronucleation, which could be observed at a value of ∼0.2 at 1.5 nM paclitaxel concentration, and mitotic arrest, which required a value of 0.6–1.0. This arrangement was true across both cell lines and all four medications. Previous research has demonstrated that paclitaxel substoichiometric binding reduces polymerization kinetics [135]. Previous studies have also shown that paclitaxel causes chromosomal segregation abnormalities and micronucleation at far lower doses than it causes mitotic stoppage [136,137]. However, ectopic spindle pole formation [137] and ectopic cleavage furrow assembly (also unknown) likely play roles in micronucleation [138]. At the highest inferred site occupancies (>0.95) and the highest external drug concentrations we examined, mitotic arrest diminished. However, because this model assumes that binding sites are uniform, it is possible that extremely high drug concentrations could access alternative, lower affinity sites, therefore these high site occupancy estimates should be regarded with caution (e.g., on exposed plus ends). The stability of syntelic kinetochore attachments at paclitaxel concentrations above 1 μM is responsible for the shortening of the duration of mitotic arrest at these levels [139]. Mitotic arrest was only reduced at paclitaxel concentrations that exceeded acceptable plasma concentrations, so this bell-shaped dosage response is fascinating but probably not relevant to chemotherapy [133,137]. A dose of 10–40 mg/kg of paclitaxel is necessary to induce tumor regression in mice, and this ranges from tumor to tumor and vehicle to vehicle [140,141,142]. After determining the paclitaxel dose necessary to cure a sensitive xenograft tumor model, Pineda et al. evaluated site occupancy in tumor cells 24 h later and found a 50% reduction in the SirTub signal at the curative 30 mg/kg dose. Thus, in the presence of 30 mg/kg of paclitaxel, SirTub occupancy would drop to below 0.1, freeing up to 90% of taxane sites for paclitaxel to bind. This is consistent with SirTub occupancy being less than 0.2 in vivo when no drug is present. Moreover, in this model, an initial paclitaxel-site occupancy of ∼0.8 was required for cure after a single dosage. What does this have to do with people who are sick? Twenty hours after paclitaxel infusion, the researchers measured the quantities of paclitaxel in the patients’ plasma and tumors. The concentrations were measured at 1.1–9 μM and 80–280 nM, respectively [137]. It is challenging to make sense of intratumoral concentrations, which measure the total of free, specifically bound, and nonspecifically bound medication. This led to the conclusion that drug efflux pumping was not a big concern and that the plasma concentration was like the extracellular concentration. To achieve site occupancy greater than 0.8, extracellular paclitaxel at 80–280 nM is required [143]. An increased mitotic index after 24 h post-paclitaxel in human malignancies is consistent with [143], this concentration being in the regime that promotes mitotic arrest [144]. As initially arrested cells reach the micronucleated interphase and as newly entering mitotic cells experience reduced drug concentrations due to clearance, micronucleation is likely to predominate at later time points. Table 2 provides a summary of several recent studies on high and low concentrations, and the impact of these levels on taxanes.

Taxanes cause mitotic delays with high site occupancy, leading to chromosomal incompatibility and micronuclei formation, although not appreciably lengthened at lower site occupancy [143]. Chromosomes, which are contained inside nuclear membranes and hence are not a physical part of the nucleus, produce them. DNA damage [152,153] and stimulation of macrophages and innate immunity through the cGAS/STING pathway may result from the development of these micronuclei due to membrane abnormalities and the loss of nuclear envelope integrity. Even modest dosages of taxanes might trigger innate immunity and inflammation by triggering micronucleation and cGAS/STING signaling. Even though the immunological checkpoint could be against the antitumor immune surveillance triggered by taxanes, this would result in fewer adverse effects, such as neutropenia and lymphopenia, than traditional therapy [154]. Upon Paclitaxel treatment, cells skip the mitotic latency and enter mitosis directly, which, like the response to DNA-damaging therapies, may activate the cGAS/STING pathway, leading to an inflammatory response at high taxane concentrations. As a result, cells that can evade mitotic death owing to taxane-based treatment are attacked by the immune system in an antitumor response. While it slows down the bone marrow’s capacity to manufacture new cells, Paclitaxel causes breast cancer cells to produce more interferon-β (IFN-β), and taxane treatment frequently causes more immune cells to enter tumors [155,156]. Paclitaxel also binds to Toll-like receptor 4 (TLR-4), activating TLR-4 downstream signaling pathways that lead to the synthesis of many proinflammatory mediators during inflammation, some of which are crucial in carcinogenesis and tumor growth. It has been shown that TLR-4 activation promotes cancer aggressiveness and chemoresistance in ovarian cancer through MyD88-dependent and-independent pathways by increasing the production of pro-inflammatory cytokines and their receptors. Thus, paclitaxel may either directly stimulate macrophages to destroy cancer cells or indirectly stimulate dendritic cells, natural killer cells, and tumor-specific cytotoxic T lymphocytes to mount an immune response against cancer. Paclitaxel suppresses regulatory T cells (Treg) and regulates myeloid-derived suppressor cells [157].

While Paclitaxel administration causes cell cycle arrest during mitosis, other evidence suggests that it induces apoptosis through numerous pathways, including the activation of signaling molecules and transcriptional activation of many genes, activation of the mitotic spindle assembly checkpoint, c-Jun N-terminal kinase/stress-activated protein kinase (JNK/SAPK) route, serine/threonine kinase-dependent phosphorylation of Bcl-2, and the p53 pathway [158]. It also activates death receptors, which in turn involves the Fas-associated death domain and caspase-8, cleaving Bid, causing mitochondrial damage and apoptosis [159]. Several important apoptosis-related proteins, including p53, polo-like kinase 1 (Plk1), caspase-2, Bim (Bcl-2 Interacting Mediator of cell death), and MAPKs ERK and JNK (extracellular signal-regulated kinase and c-Jun amino-terminal kinase) are activated by paclitaxel in breast carcinoma cells; however, Bcl-2 (highly phosphorylated upon taxane treatment) appears to be the most important protein, as its silencing delays mitosis and reduces cell death [160]. Leukemia cells become more resistant to paclitaxel-induced apoptosis and mitochondrial dysfunction when Bcl-2 phosphorylation is abolished [158], which is a consequence of taxane therapy inducing bcl-2 phosphorylation. Finally, Paclitaxel treatment promotes death receptor (DR4 and DR5) upregulation in melanoma [161]. 

Histopathological studies have highlighted the anti-angiogenic effects of taxanes, along with their anti-mitotic activity, by studying their effect on the development of capillary tubes and microvessel density [162]. Intriguingly, a reduction in microvascular density is caused by a modest dosage of paclitaxel and DTX (docetaxel), which inhibit the accumulation of bone marrow-derived endothelial progenitor cells (EPC) in the tumor microenvironment [163]. The formation of ROS, including hydrogen peroxide and superoxide in normal and neoplastic cells, plays a major role in the regulation and induction of several molecular processes, such as apoptosis, cell proliferation, survival, and treatment resistance [164]. ROS production may cause stress and damage cells via oxidative stress [165]. The balance between their production and removal is critical for cancer cell survival, and cancer cells use a variety of mechanisms to mitigate their toxic effects. For instance, ROS accumulation can cause constitutive activation of transcription factors such as nuclear factor κB (NF-κB), activator protein AP-1, hypoxia-inducible factor 1 (HIF-1), and signal transducer and activator of transcription 3 (STAT3), which in turn promotes cell proliferation through the aberrant activation of ERK1/2 or inhibition of tyrosine phosphatase [164]. Therefore, altering ROS levels in cells and the molecular pathways connected to ROS may be a viable method for combating multi-drug resistant tumors as it may be used to trigger cancer cell death or boost the efficacy of chemotherapeutic drugs. For instance, paclitaxel administration produces a considerable generation of mitochondrial ROS, which activates the STAT3 signaling pathway [166]. Uncoupling protein 2 (UCP-2) is a protein found in the inner mitochondrial membrane that decreases ROS generation in the presence of oxidative stress. High levels of UCP-2 expression in A549 and H460 lung cancer cell lines inhibited paclitaxel-induced ROS generation and STAT3 activation [167]. These two tumor cell lines showed dramatically enhanced cell death after paclitaxel treatment when UCP-2 was downregulated [166]. Interestingly, recent advances have resulted in the creation of paclitaxel-loaded ROS-sensitive polymer-based nanoparticles (NPs) that can selectively release their cargo in tumor tissues, which are often characterized by high levels of ROS. This type of NP accumulates in the liver, spleen, and lungs, which may make it useful for targeting distant cancer metastases [168]. 

### 2.8. Camptothecin (CPT)

The first natural chemical extracted from *Camptotheca acuminata* was CPT. It was initially produced in 1966 and is a quinoline alkaloid [169]. Topotecan, a camptothecin (**CPT**) derivative, has been utilized as both a frontline and secondary chemotherapy treatment for small cell lung cancer (SCLC). It is the only medicine approved for use as a second-line chemotherapeutic treatment for recurrent SCLC in the European Union and the United States. Combinations of cisplatin and irinotecan have been used for the treatment of SCLC [170]. Numerous in vitro and in vivo studies [171,172] have shown that camptothecin (**CPT**) and its derivatives, including irinotecan (CPT-11, 4), belotecan (CKD-602, 5), and 10-hydroxycamptothecin (HCPT), exert a broad spectrum of anti-tumor activities against a variety of tumor types, including ovarian cancer, NSCLC, and refractory colorectal cancer. In Korea, patients with non-small cell lung cancer (NSCLC) or ovarian cancer may now be treated with belotecan, a relatively novel camptothecin (**CPT**) derivative. It has been reported to be as effective as previous camptothecin (**CPT**) formulations while being less toxic [173]. Multiple camptothecin (**CPT**) analogs are in different phases of clinical development [174], including 7-(4-methylpiperazinomethylene)-10, 9-aminocamptothecin, exatecan mesylate, 11-ethylenedioxy-20(S)-camptothecin, 9-nitrocamptothecin, and karenitecin. However, they have a host of unpleasant side effects such as diarrhea, weakness, myelosuppression, stomatitis, nausea, vomiting, stomach discomfort, hair loss, and peripheral neuropathy [175].

Camptothecin (**CPT**) inhibits the growth of a variety of malignancies via different mechanisms and has now moved quickly into clinical trials (Figure 3). Camptothecin (**CPT’s**) anticancer activities may be traced back to its ability to inhibit topoisomerase-1 (Topo I) at the molecular level [176]. Structural models demonstrate that camptothecin (**CPT**) binds to the Topo I-DNA binary complex in a non-covalent manner. Structure-activity correlations provide insight into a possible mechanism by which camptothecin (**CPT**) and its derivatives decrease Topo I expression [177]. While camptothecin **(CPT**) has strong anti-neoplastic effectiveness and an original mode of action, it also has several undesirable characteristics that limit its therapeutic use. Camptothecin (**CPT**) is difficult to administer owing to its relatively poor water solubility. In addition, the open form of camptothecin (**CPT**) carboxylate is produced when the -hydroxy lactone ring (ring E) opens under physiological conditions. While carboxylic acid and its sodium counterpart are both soluble, their anticancer potential is greater than that of camptothecin (**CPT**). In addition, albumin binds well to this ionic form, thereby decreasing the available drug concentration [178]. While camptothecin (**CPT’s**) anticancer capabilities sparked a great deal of interest in studying it, it has also been studied for other features, such as its insecticidal and antiviral effects. While semisynthetic camptothecin (**CPT**), such as irinotecan and topotecan maintain their place in chemotherapy, several additional anticancer drugs have been produced and are in various stages of preclinical testing or clinical research [179]. 

Clinical trials involving camptothecin (CPT) were put on hold for the time being, although investigation into the compound’s mechanism of action has proceeded. Many anticancer medications work by damaging deoxyribonucleic acid (DNA) strands in cancer cells while leaving healthy ones intact. Camptothecin’s (CPT) topoisomerase I (Top I) inhibition is the key to its anticancer activity, which was discovered in the 1980s [182]. DNA replication requires the latter. More specifically, camptothecin (CPT) was discovered to inhibit the Top1-DNA complex, as opposed to the free Top1 enzyme. The discovery of this molecular target of camptothecin (CPT) has led to potential developments in the chemistry and SAR of camptothecin (CPT), permitting the manufacture of analogues with enhanced potency, superior selectivity, and consequently reduced toxicity [183]. In order to achieve selectivity, it is necessary to target cells with a higher expression of the TOP I enzyme, which is primarily found in cancer cells [184]. Those with a higher concentration of camptothecin (CPT) in their systems are more sensitive to it [185]. Accordingly, the cytotoxic impact of camptothecin (CPT) is enhanced in cells with elevated levels of Top I. Camptothecin (CPT) has several undesirable side effects during clinical use. These include nausea, stomach cramps, diarrhea, and even hemorrhagic cystic illness [185]. 

Camptothecin’s (CPT) target enzyme, Top1 Top I, elucidated key aspects of the drug’s mechanism of action. Supercoiling, caused by DNA and RNA polymerases, occurs in the DNA double helix during replication [186]. Top1 regulates DNA topology, which alleviates supercoiled DNA [187]. It cuts the supercoiled part of the DNA, resulting in a single-strand break, and interacts with the DNA phosphate backbone via a phosphotyrosine link. Then Top I bind covalently to the snipped 3′ end. This permits the 5-nicked strand to unwind and rotate around the unbroken strand, followed by Top I catalyze the inverse response by re-ligation of the cut strand, easing the supercoil’s torsional tension. Since Top I is directly implicated in these processes, it follows that it also plays a role in recombination, transcription, and DNA repair [188].

In contrast to Top I, Topoisomerase 2 (Top2) does not have activity against CPT, and when it breaks supercoiled DNA, it creates a double strand break rather than a single damage [189].

Camptothecin (CPT) primarily targets the Top I and DNA complex, also known as the “Top I covalent complex” [185]. The addition of camptothecin (CPT) creates a ternary complex. However, camptothecin (CPT) strongly shifts this equilibrium towards the formation of the ternary complex, reducing the amount of free Top I, and eventually inhibiting its effect [185], under normal physiological circumstances, the equilibrium between unbound Top I and the Top I-DNA complex shifts toward the free enzyme. Many researchers believe that the ternary complex impedes the progress of the DNA replication fork. Camptothecin (CPT) forms hydrogen bonds with both Top I and DNA, blocking both the reation of the nicked DNA and the release of Top I from the DNA [190]. As a result, DNA strand breaks build up, and cancer cells commit suicide during the S phase of the cell cycle [190]. Top I inhibition specifically makes the DNA supercoil more noticeable, which in turn inhibits RNA and DNA polymerase [190].

This will cause DNA damage and a block in RNA (including ribosomal RNA) production. Due to this, cell division stops, and apoptosis sets in. In this way, camptothecins (CPTs) are an example of an interfacial inhibitor since they reversibly trap macromolecular Topoisomerase I-DNA complexes and have a single biological target (Topoisomerase I).

### 2.9. PTOX 

PTOX is a widely occurring, structurally diverse lignan that is classified as arylnaphthalene. Traditional Chinese remedies rely heavily on PTOX derivatives (PTOXs, sourced from *Dysosma versipellis*, *Diphylleia sinensis*, and *Sinopodophyllum hexandrum*) for their therapeutic effects. The first-line therapy for condyloma acuminatum (anogenital warts) has been 0.5% PTOX since it was approved for use by the World Health Organization in 1990. Two major semi-synthesized PTOX glycosyl derivatives, etoposide (VP-16) and teniposide (VM-26) which are topoisomerase inhibitors were authorized by the FDA in 1983 and 1992, respectively. A variety of cancers, including SCLC and testicular cancer, respond to VP-16 when administered as a first-line treatment [191]. For acute lymphoblastic leukemia, the drug of choice is VM-26 [192]. However, because of their selective nature toward rapidly replicating cells, VP-16, VM-26, and other PTOX-derived medications often exhibit substantial side effects, including suppression of bone marrow, hair loss, and neurotoxicity. In addition to cancer cells, many other types of rapidly proliferating normal cells, such as bone marrow and hair follicle cells, are at risk. New PTOX-derived compounds also induce drug resistance, resulting in serious adverse effects such as metabolic dysfunction and secondary carcinogenesis [193,194]. More effort has been put into creating effective drug discovery based on PTOX structural alterations, clinical guidance, and a wide variety of partner medications to overcome the limitations of current medical practices. 

Several applications of PTOX derivatives have been discovered in recent years. For some reason, the cis-lactone form of picropodophyllotoxin (PPP) has no impact on microtubules and is not cytotoxic [195]. PPP does not lead to DNA damage, such as etoposide and PTOX, because it does not inhibit topoisomerase II or tubulin. Despite these findings, PPP has been mostly ignored. The significance of insulin-like growth factor-1 receptor (IGF-1R) tyrosine phosphorylation in the transformation and proliferation of malignant cells was not fully understood until 2004. In contrast to IGF-1R, whose beta subunit is 95 kDa, the insulin receptor (IR) was resistant to inhibition by PPP [196]. IGF-1R confers protection to tumor cells with a malignant phenotype against antitumor therapy [197]. However, IGF-1R is not strictly required for regular cell development. By significantly inhibiting IGF-1R activity, PPP was able to decrease pAKT and phosphorylated extracellular signal-regulated kinases 1 and 2 (pERK1/2), induce apoptosis, and lead to complete tumor regression in xenografted and allografted mice. IGF-1R autophosphorylation may be inhibited at the substrate level by PPP, which was found to not affect the insulin receptor and did not compete with ATP in an in vitro kinase assay [198]. In 2010, PPP was shown to lower IGF-1R and AKT phosphorylation and impede the development of human glioma cell lines. PPP can cross the blood-brain barrier in vivo [199,200] and has been shown to significantly reduce tumor growth in mice after subcutaneous tumor transplantation. Furthermore, mouse glioma stem cells induced by radiation at intervals induce radiation resistance by upregulating IGF-1R expression. While these tumors are resistant to radiation in vivo, PPP monotherapy drastically reduces their growth. Surprisingly, PPP therapy also increases radiation sensitivity in cancers. In a study, independent of the insulin-like growth factor-1 receptor (IGF-1R) pathway, it interfered with microtubule dynamics to decrease cancer. There is no evidence of neurotoxicity with PPP, and the only adverse effect is neutropenia, which may be reversed [201,202,203,204].

There is some evidence that podophyllotoxin may be effective against cancer. Precursors of this drug show great promise in targeting pre-metastatic cells, giving it an attractive option for the treatment of a wide range of malignancies, including leukemia, testicular, prostate, lung, and ovarian [205].

DNA replication is slowed when podophyllotoxin binds to topoisomerase II, demonstrating its cytotoxic effects. A high concentration of podophyllotoxin reduces the amount of topoisomerase II. Due to their strong binding, etoposide and podophyllotoxin cause more damage to the DNA duplex, which in turn raises the risk of DNA damage in mammalian tissues where double-stranded DNA is present. Damage to DNA may lead to cell death in many ways, including insertions, deletions, and recombination, all of which contribute significantly to the overall amount of DNA damage (Figure 4) [206].

### 2.10. Additional Anticancer Drugs Derived from Plants 

The hydrophobic ester ingenol mebutate (IM) is derived from the diterpene ingenol, which was first discovered in the common Australian shrub, *Euphorbia peplus* (Euphorbiaceae). Actinic keratosis, a common skin disease caused by persistent UV radiation exposure that may develop into squamous cell carcinoma if left untreated, is a condition for which IM has been approved as a topical therapy. High doses (200–300 μM) cause fast induction of cell death in the treated region, whereas low concentrations (0.1 μM) stimulate an inflammatory response that may clear out any lingering cells. Detailed discussions on ingenol mebutate pharmacology, mechanism of action, pharmacokinetics, dosage, and routes of administration have been compiled [96]. Among other drugs, homoharringtonine (HHT) is an authorized medication for chronic myeloid leukemia. It is a naturally occurring ester of the alkaloid cephalotaxine obtained from trees of the genus Cephalotaxus (Cephalotaxaceae) [207]. HHT limits protein synthesis by binding to the A-site cleft of the large ribosomal subunit, which disrupts chain elongation [208]. Patients with resistance and intolerance to hypomethylating drugs such as azacitidine and decitabine [209] have been successfully treated with a semi-synthetic form of HHT known as omacetaxine mepesuccinate for myelodysplastic syndromes (MDS) and chronic myelomonocytic leukemia (CMML). Cape bushwillow, or *Combretum caffrum* (Combretaceae), is a plant native to South Africa and is the source of a group of cis-stylbenes known as combretastatins. Compounds of the combretastatin family block tubulin polymerization, which has a knock-on effect on cancer cells by disrupting the tumor endothelial cells lining the tumor vasculature and triggering a rapid vascular collapse in solid tumors [83,210]. The two chemicals found in nature are combretastatin A1 and A4 [211]. The orphan medication combretastatin A4 phosphate (CA4P) has been licensed by the FDA for the treatment of several types of thyroid and ovarian cancers [212].

### 2.11. Inflammation vs. Cancer 

One-fourth of all malignancies are associated with chronic inflammation or infection [213]. Prominent regulators of cancer-related inflammation include NF-κB, nuclear factor of activated T cells (NFATc), HIF-1, and STAT3 [214,215]. Natural products have been proven to reduce the pro-inflammatory and cancer-proliferative effects of these indicators. The NF-κB and STAT3 pathways are mostly activated after repeated exposure to topical carcinogens or UVB radiation, which leads to the development and progression of skin malignancies [216,217]. COX-2 is increased after UVB exposure [216], and converts arachidonic acid to prostaglandin E2 (PGE), a key step in the activation of the pro-inflammatory cascade. Many studies have investigated the effects of ginger on skin cancer. For example, one such study used a two-stage mouse skin carcinogenesis model in which 12-O-tetradecanoylphorbol-13-acetate (TPA) was injected into the mice to assess the antitumor promoting activity of [6]-gingerol, a main pungent component of ginger [218]. Pretreatment with 6-gingerol lowered tumor size and vascular permeability, both of which are signs of decreased inflammation [218]. Interestingly, in the same study, curcumin, which also reduces epidermal permeability, followed a similar pattern [218]. Similarly, researchers discovered that topical administration of 6-gingerol before TPA treatment decreased inflammation and TNF levels [219]. According to Wu et al. [220], 6-shogaol, one of the constituents of ginger, suppresses tumor promotion and inflammatory markers following TPA administration by reducing the expression of inducible nitric oxide synthase (iNOS) and cyclooxygenase 2 (COX-2) and blocking pathways upstream of NF-κB and activator protein 1 (AP-1). Both 6-gingerol and curcumin inhibit iNOS and COX-2 expression [220]. These studies suggest that topical administration of ginger components may help prevent or reduce inflammation caused by skin cancer. 

The anti-inflammatory effects of EGCG, a green tea polyphenol, on skin cancer have been the subject of several studies. A reduction in skin tumor formation was observed in mice treated topically with EGCG post-exposure to UVB light, with NF-kB suppression proposed as a mechanism for this protective effect [221]. Furthermore, AP-1, a transcription factor implicated in both inflammation and tumor development, is downregulated by EGCG [222]. Green tea polyphenol-treated volunteer skin was examined for erythema, and DNA damage (represented by cyclobutene pyrimidine dimers) was measured before and after UVB exposure. Lower levels of these dimers were seen in both the dermis and epidermis post-treatment with green tea phenols [223]. The probability of developing skin cancer after exposure to UVB decreased with increasing topical dosages of green tea polyphenols [223], suggesting that green tea extracts may help treat skin cancer symptoms. To further back up this finding with solid evidence, a clinical trial (NCT02029352) is currently underway, assessing the use of topical green tea for treating superficial skin cancer in patients with basal cell carcinoma through the Maastricht University Medical Center. 

The acute inflammatory response stimulates the innate and adaptive immune responses and is the body’s first line of defense against external infection or damage. Hematopoietic cells that have undergone evolutionary diversification and are part of the innate immune system include neutrophils, macrophages, dendritic cells, mast cells, and so on [224]. It is well established that these cell subsets aid in the resolution of inflammation by phagocytizing pathogens, germs, and necrotic material. DCs and macrophages, in their roles as antigen-presenting cells, have been demonstrated to deliver antigens to T cells, allowing the latter to recognize and activate the adaptive immune response [225]. This suggests that the acute inflammatory response may be useful in fighting off infections by eliminating germs. The presence of many immunosuppressive cells (M2 macrophages, MDSCs, Treg cells, etc.) and cytokines [226,227] indicates that the acute inflammatory response has not resolved in a timely manner, and that chronic inflammation has developed consequently. There is evidence that these alterations enhance oncogene activation, DNA and protein damage, ROS production, and the downregulation of tumor suppressor genes such p53 and NF-kB [226]. DNA methylation, histone modification, chromatin remodeling, and noncoding RNA are all examples of epigenetic modifications that play a role in cancer’s initiation, progression, invasion, metastasis, and medication resistance [228,229,230,231]. Macrophage histone lactylation, which may aid in inflammation resolution and tumor immune escape, [229,232,233,234,235] deserves special note. Whether or how lactylation can affect other proteins and what such changes might do to the proteins’ activities is still up for debate. In addition, lactic acids in the inflammatory microenvironment have been shown to act on immune cells (including cytotoxic T cells (CTLs), DCs, and APCs) [236,237,238] and immunosuppressive cells (including M2-macrophages, MDSCs, and Treg cells) [239,240,241] to promote the development of inflammation and cancer. Gene abnormalities would cause aberrant cell multiplication, but the immune system could distinguish tumor cells based on their unique antigens and eliminate them. Inflammatory factors and signaling pathways, such as 5-LOX, COX-2, TGF-β, and VEGF are well-known to connect inflammation and chronic illnesses [230]. Furthermore, aberrant inflammatory pathways such NF-κB, MAPK, JAKSTAT, PI3K/AKT, etc., often lead to the dysregulation of inflammatory molecules or factors (Figure 5). For instance, the NF-B signaling system regulates the expression of about 500 genes involved in cancer [242]. 

The role of inflammation in the initiation, maintenance, and dissemination of cancers is well established [243]. It has been suggested that chronic inflammation, which is fueled by immune cells and molecular signaling pathways, makes people more prone to developing cancer. Up to 25% of malignancies have been linked to chronic inflammatory illnesses, although the mechanism linking the two is still unknown [231]. Precancerous tumor lesions have been linked to certain chronic inflammatory disorders. Inflammatory bowel disease (IBD) is a well-known risk factor for colorectal cancer (CRC). Clinical evidence suggests that IBD may lead to the development of cancerous tumors over the course of many decades. In addition, it is well-documented that pharmacological stimulation of IBD is a standard approach for triggering CRC in mice [244,245]. DNA methylation levels are different in IBD-related colon cancer vs. random colon cancer [246]. Human CRC single cell multiomics sequencing studies have shown that epigenetic inheritance plays a critical regulatory role in the onset and progression of CRC [247]. Several chronic inflammatory disorders caused by viruses and bacteria have been linked to carcinogenesis. It has been shown that H. pylori infection causes gastritis and stomach cancer [248,249]. While inflammatory bowel disease alone is not sufficient to generate colorectal cancer, [250] colonic inflammation and colitis-associated colon carcinogenesis may be exacerbated by the use of a common antibacterial addition in mice [251]. Chronic hepatitis, which may lead to primary HCC, was also linked to HBV infection [252,253]. Infection with human papillomavirus (HPV) is an established risk factor for developing cervical cancer [254]. Additionally, it has been demonstrated that gut microbiota can affect immune cell differentiation and function directly or indirectly (via metabolites like polysaccharide-dextran, LPS, deoxycholic acid (DCA), short-chain fatty acids (SCFA), butyrate, and propionate) on cancer. Examples of immune cells affected include M2 macrophages [255], neutrophils [256], Treg cells [257,258,259], T cells [260], and NKT cells [261]). The effectiveness of immunotherapy was significantly enhanced by treatments aimed at the gut microbiota [262]. Tumorigenesis has also been linked to certain chronic autoimmune disorders [263]. Furthermore, numerous immunosuppressive cells limit the killing activity of T cells in the chronic inflammatory milieu, leading to immune escape and increasing tumor growth. There is mounting evidence that persistent inflammatory stressors raise the cancer risk, advance tumor growth, and facilitate metastasis [243]. In this way, the inflammatory cells and cytokines generated during chronic inflammation may serve as tumor promoters by influencing cell survival, proliferation, invasion, and angiogenesis.

### 2.12. Colorectal Cancer 

The Centers for Disease Control and Prevention (CDC) defines colorectal cancer as the unregulated growth of cells in the colon or rectum. According to recent evidence, the prognosis for colorectal malignancies is bleak when systemic inflammation is present [264]. In addition, elevated blood CRP levels are seen in 21–41% of patients with a resectable illness, showing systemic inflammation in response to the tumor [264]. Inflammation aids the spread and invasion of cancer cells, which may explain why it is often present in the development of colon cancer [265]. High levels of pro-inflammatory cytokines are observed in patients with colitis-associated cancer, a type of cancer that may be traced back to chronic inflammation [266]. These cytokines can cause mutations in oncogenes and tumor suppressor genes, which in turn promote the development of illness. Constant activation of NF-κB and STAT3 mediates the immunological response and oncogenesis [266] in many cases of colon cancer. Several studies have examined the role of curcumin in colon cancer. According to a study by Marjaneh et al., curcumin lowered chronic inflammation and reduced disease activity in mice [267]. In another study, piperine suppressed mTORC1 activity [268], which is linked to inflammation and cancer development, both with and without curcumin. Curcumin in combination with piperine exhibited anti-inflammatory characteristics in colon cancer, as did individual phytochemicals and combinations of phytochemicals that lowered TNF- and COX-2 levels. Patients at high risk of developing colorectal neoplasia were included in a phase IIa clinical study of curcumin [269]. Administration of 4 g of curcumin orally daily for 30 days significantly reduced aberrant crypt foci in patients with colorectal cancer [269]. In another study, patients at a high risk of getting colorectal cancer were studied using ginger supplements [270]. Interestingly, patients on ginger supplements had lower levels of MIB-1 and telomerase reverse transcriptase expression [270]. The inflammatory cytokine MIB-1 was upregulated in the gastrointestinal mucosa of guinea pigs in a study that mimicked mucosal inflammation [271]. Thus, gingerol shows the potential to reduce inflammation and slow disease progression in patients with colorectal cancer. The powerful anti-inflammatory effects of another phytochemical quercetin in the colon were demonstrated by the fact that quercetin-treated rats who were administered with azoxymethane injections showed reduced levels of inflammatory markers compared to saline-treated rats [272]. Inhibition of the NF-κB pathway in colon cancer cells was identified as the mechanism by which quercetin exerted anticancer properties, as reported by Zhang et al. [273]. Research by Han et al. confirmed the ability of quercetin to reduce NF-κB and TLR-4 expression [265]. This results in reduced colon cancer cell migration and invasion. Their study results also showed a drop in the levels of TNF-α, COX-2, and IL-6, all inflammatory markers [265] (Figure 6). 

### 2.13. Prostate Cancer 

According to the American Cancer Society’s estimates, prostate cancer is the second most common cancer in men. A blood test for prostate-specific antigen (PSA) helps to detect the disease. Even though a high PSA level is not very sensitive, it suggests prostate inflammation [274]. Carcinogenesis and tumor progression may be influenced by persistent inflammation of the prostate [275]. The tumor microenvironment is driven by angiogenesis and the epithelial-mesenchymal transition in the presence of chronic inflammation [276], which in turn leads to disease development and metastasis in the prostate. Treatment results are strongly influenced by the level of chronic inflammation in the tumor microenvironment. TNF-α, TGF-β, IL-7, IL-2, and macrophage inflammatory protein-1b [266] are all pro-inflammatory mediators that have been linked to prostate cancer. Prostate cancer-related inflammation can be mitigated by curcumin treatment. Both C-X-C motif ligand 1 (CXCL-1) and C-X-C motif ligand 2 (CXCL-2) are chemokines that influence prostate cancer cells, and curcumin therapy greatly lowers the transcription of both chemokines [277] by blocking NF-κB activation by preventing the phosphorylation of NF-κB inhibitor and downstream phosphorylation of P65. Pomegranate, green tea, broccoli, and turmeric were mixed into a single capsule and tested in a human investigation of prostate cancer, in one study [278]. While PSA levels increased in both the treatment and placebo groups, they increased less in the treatment group, suggesting that polyphenol-rich diets may be helpful in prostate cancer treatment [278]. Supplementation with curcumin leads to greater endogenous antioxidant capabilities, as shown by studies showing reduced superoxide dismutase activity and increased plasma total antioxidant capacity in prostate cancer male patients undergoing radiation [279] (Figure 6). 

## 3. Conclusions 

In the current manuscript, we described the anticancer and anti-inflammatory activities of a group of phytochemicals and focused on the interrelationship between cancer and inflammation. As confirmed by numerous investigations, phytochemicals represent optimum pharmacological activities with minimal adverse effects. Additionally, they modulate multiple cellular signaling pathways to exert their specific anticancer effects. It is believed that high dietary consumption of fruits and vegetables reduces the risk of cancer onset.

Several signaling pathways are involved in the development of cancer, and phytochemicals can target these signaling pathways with the aim of multitargeted therapies. We have described the most important molecular targets of phytochemicals in this review article. The synergic and additive activity of certain phytochemicals could be combined with chemotherapy, limiting the doses and frequency of the chemotherapeutic drugs, and minimizing the toxicity.

## Figures and Tables

**Figure 1 ijms-23-15765-f001:**
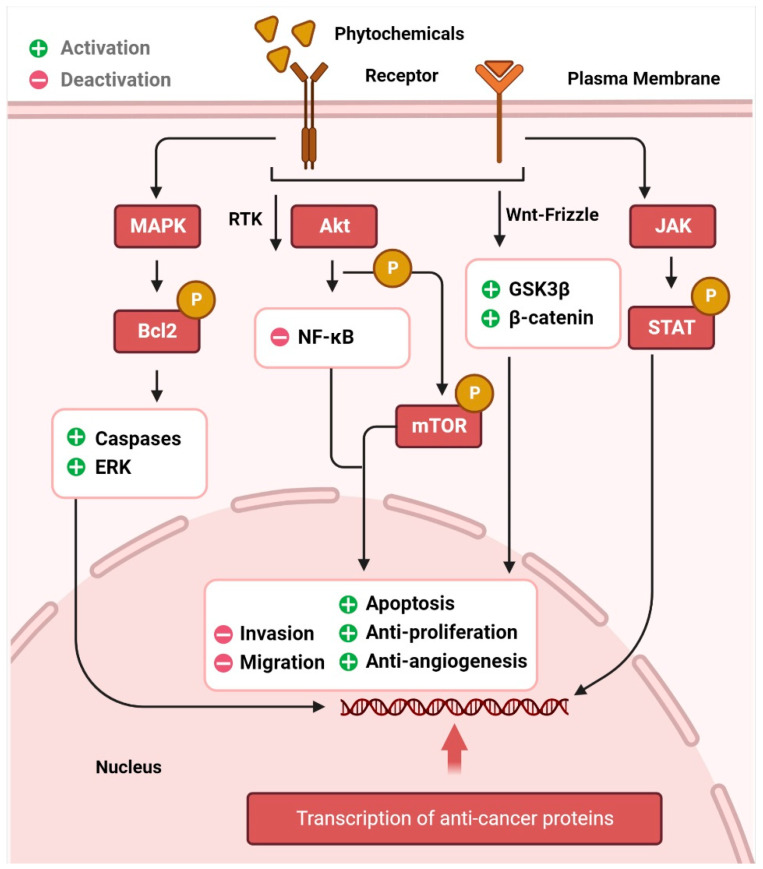
Cancer chemoprevention through the phytochemical interaction with signaling molecules. The image illustrates how phytochemicals activate MAPK, Akt, Wnt, and JAK/STAT pathways, leading to cancer cell death through many intracellular signaling molecules.

**Figure 2 ijms-23-15765-f002:**
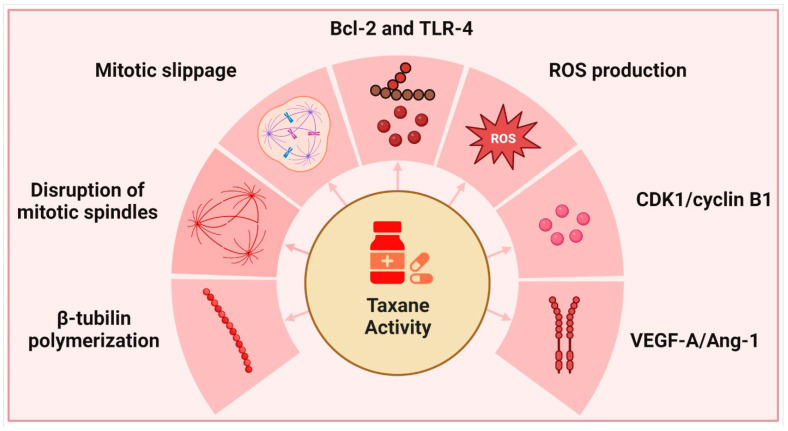
Taxanes and their primary modes of action.

**Figure 3 ijms-23-15765-f003:**
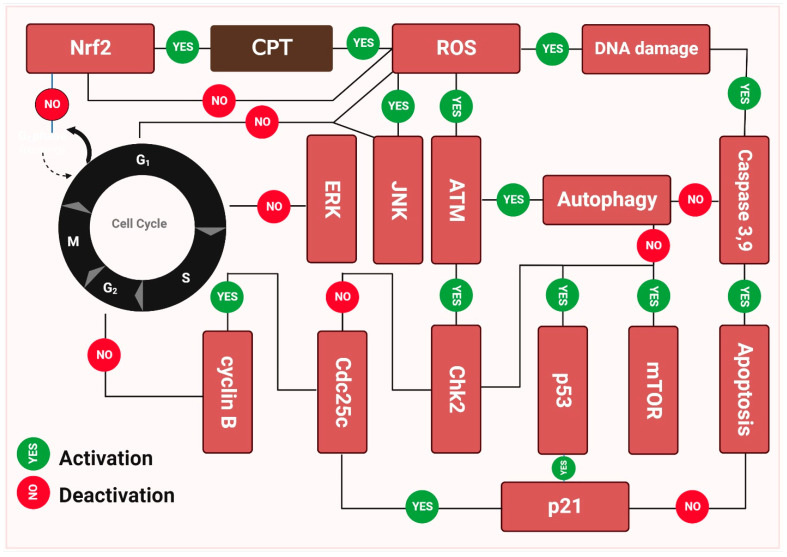
The molecular basis behind camptothecin (CPT’s) antitumor activity. The capacity of camptothecin to bond with double-stranded DNA is a major factor in its anti-tumor effect. During the replication process, these enzymes nick the double-stranded DNA, releasing a single-stranded copy. In addition, they significantly minimize supercoiling in DNA double helix structures. Camptothecin inhibits cancer growth by inhibiting topoisomerase-I catalytic activity after a non-covalent binding, leading to elevated levels of p21, p53, and mTOR expression. As opposed to this, camptothecin stimulates ERK and NERF2, both of which are upregulated by the drug. The cell death seen in Figure 3 was the result of these molecular processes, which triggered apoptosis [180,181].

**Figure 4 ijms-23-15765-f004:**
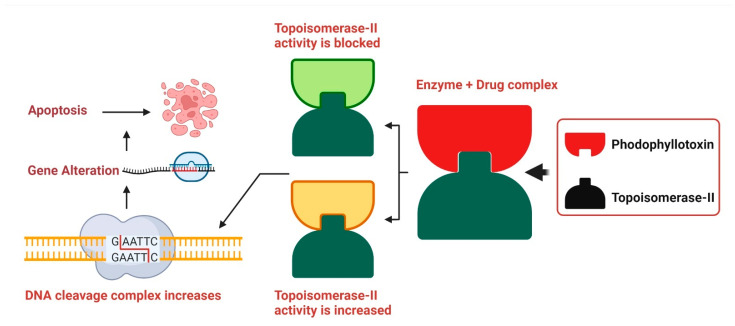
Podophyllotoxin’s Mode of Action. The DNA replication inhibitors etoposide and teniposide interact with topoisomerase II, and podophyllotoxin is a suitable replacement for them. DNA cleavage is accelerated by this bioactive compound because it stimulates the production of topoisomerase II. Though it does not block topoisomerase II’s catalytic activity, etoposide is toxic to the enzyme, resulting to increased DNA duplex cleavage and irreparable double-stranded DNA breaking. However, genetic rearrangements such recombination, translocation, deletion, and insertion lead to cell death. Originally published by Kumar et al. [206].

**Figure 5 ijms-23-15765-f005:**
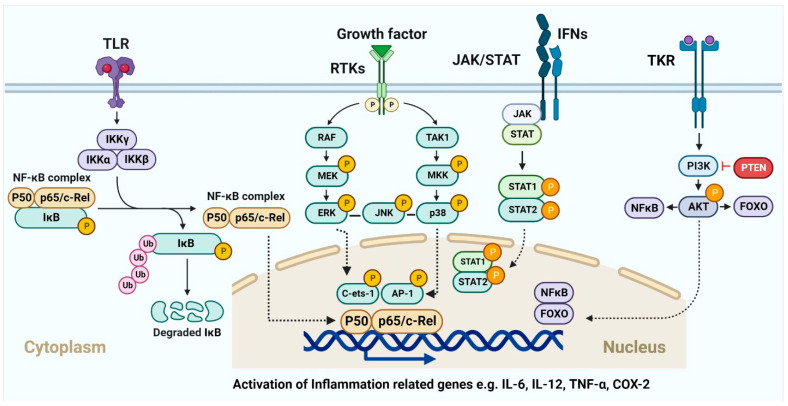
Inflammation and tumor progression mechanisms involving signaling pathways. Various membrane receptors trigger intracellular signaling pathways that contribute to processes including inflammation and tumor growth. Numerous well-understood pathways, including NF-κB, MAPK, JAK-STAT, and PI3K-AKT, are activated by subsequent downstream signaling events. These mechanisms control a wide range of inflammatory variables.

**Figure 6 ijms-23-15765-f006:**
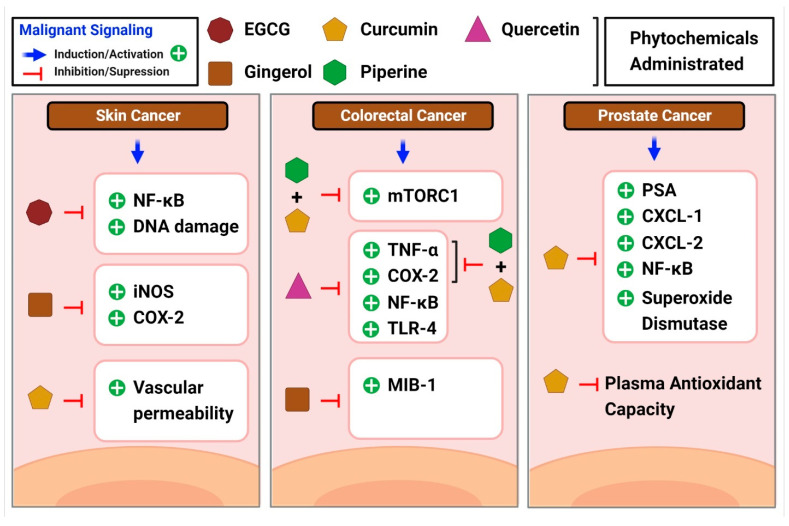
A brief overview of the common inflammatory markers linked to different cancer types and moderating effects shown by phytochemicals in the diseases.

**Table 1 ijms-23-15765-t001:** Major classification of phytochemicals and their uses in cancer treatment.

Classes and Subclasses of Phytochemicals	Therapeutic Pharmacology	Distinct Cancers	Molecular Objectives	References
**Vinca alkaloids**				
Vinblastine	Preventing microtubules from turning anti-mitotic by adhering to tubulin	Non-small-cell lung carcinoma (NSCLC), breast and lung cancers, leukemia, Hodgkin’s and non-Hodgkin’s lymphomas, testicular carcinoma, melanoma, head and neck cancer, uterine melanomas, Kaposi’s sarcoma, and transitional cell carcinoma	Microtubules and tubulin	[84,85,86,87,88,89,90]
Vincristine	Mitosis inhibition through binding to tubulin dimer			
Vindesine	Anti-mitotic			
Vinflunine	Anti-mitotic			
Vinorelbine	Prevents cancer cells from beginning mitosis by blocking the transition from metaphase to anaphase; increases the rate of apoptosis			
**Taxanes**				
Docetaxel	Microtubule inhibition causing cell cycle arrest and abnormal mitosis	Prostate, head and neck, breast cancers, gastric adenocarcinoma, and NSCLC	Microtubules and tubulin	[91,92]
Cabazitaxel				
Paclitaxel				
**Camptothecin**				
Irinotecan	Protects against deadly double-stranded DNA breaks by stabilizing topoisomerase I-DNA complex and halting religation of single-strand breaks	Small cell lung cancer (SCLC), ovarian, cervical, and colorectal cancers	Topoisomerase-I	[93]
Topotecan				
**Podophyllotoxin**				
Etoposide	Forms a complex with topoisomerase II and DNA blocking the enzyme’s ability to synthesize new DNA	NSCLC, cervical, nasopharyngeal, colon, breast, testicular, osteosarcoma, and prostate cancer	Topoisomerase-II	[94]
Teniposide				
**Other phytochemicals**				
Homoharringtonine	Interferes with chain elongation and inhibits protein synthesis by binding to the large ribosomal subunit	Chronic myeloid leukemia	Ribosomal protein	[95]
Ingenol mebutate	Immediate cell death and inflammatory response activation	Actinic keratosis	Protein kinase C (PKC)	[96]
Combretastatin A4	Tumor endothelial cells are disrupted when tubulin polymerization is inhibited, resulting in loose tumor blood vessels	Anaplastic thyroid cancers	Tubulin	[97]

**Table 2 ijms-23-15765-t002:** Effects of taxanes with adequate concentrations. ↑ represents upregulation ↓ represents downregulation.

Taxanes	Cancer Type	Days/Time	Concentration	Effects	References
**Paclitaxel**	triple-negative breast cancer (TNBC)	weekly	80 (mg/m^2^)	↑ stromal tumor-infiltrating lymphocytes and micronucleation	[145]
	gefitinib-resistant non-small-cell lung cancer (NSCLC) cells (PC9-MET)	72 h	50–100 nM	↓ Sustainability and ↑ morphological signs of apoptosis.	[146]
**Docetaxel and Paclitaxel**	breast Cancer	9 weeks and 12 weeks	75–100 mg/m^2^ and 80 mg/m^2^	Hair loss (alopecia)	[147]
**Docetaxel**	breast invasive ductal carcinoma	3 weeks for 6 cycles	75–100 mg/m^2^	Upper Gastrointestinal Bleeding	[148]
**Docetaxel**	prostate adenocarcinoma	every 3 weeks	75–100 mg/m^2^	bilateral pain of the two inferior limbs and bilateral motor deficit	[149,150]
**Cabazitaxel**	metastatic castration-resistant prostate cancer (mCRPC)	q3week and q2week	25 mg/m^2^ and 16 mg/m^2^	No cumulative grade ≥3 neuropathy or nail disorder and one case of febrile neutropenia.	[151]

## Data Availability

Not applicable.

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
