# Peer review of "Phytochemicals as Chemo-Preventive Agents and Signaling Molecule Modulators: Current Role in Cancer Therapeutics and Inflammation"

_ijms, 2022, doi:10.3390/ijms232415765_

Round 1
Reviewer 1 Report
After reviewing a manuscript "Phytochemicals as chemo-preventive agents and signaling molecule modulators: Current role in cancer therapeutics and inflammation", I suggest the following corrections:
1. The manuscript is lacking in novelty. Manuscript text is a combination of articles that were already written. Role of phytochemicals, described in this manuscript (epipodophyllotoxin, camptothecin derivatives, taxane diterpenoids, vinca alkaloids, quercetin, curcumin, piperine, epigallocatechin gallate (EGCG), and gingerol) in cancer prevention are already described in details in many articles, reviews.
Some of published articles are:
a. G. MS, Swetha M, Keerthana CK, Rayginia TP and Anto RJ (2022) Cancer Chemoprevention: A Strategic Approach Using Phytochemicals. Front. Pharmacol. 12:809308. doi: 10.3389/fphar.2021.809308
b. Rahman MA, Hannan MA, Dash R, Rahman MDH, Islam R, Uddin MJ, Sohag AAM, Rahman MH and Rhim H (2021) Phytochemicals as a Complement to Cancer Chemotherapy: Pharmacological Modulation of the Autophagy-Apoptosis Pathway. Front. Pharmacol. 12:639628. doi: 10.3389/fphar.2021.639628
c. Mosca et al., Taxanes in cancer treatment: Activity, chemoresistance and its overcoming, Drug Resistance Updates, 54, January 2021, 100742
d. Ghanbari-Movahed, M.; Kaceli, T.; Mondal, A.; Farzaei, M.H.; Bishayee, A. Recent Advances in Improved Anticancer Efficacies of Camptothecin Nano-Formulations: A Systematic Review. Biomedicines 2021, 9, 480. https://doi.org/10.3390/biomedicines9050480
e. Fan HY, Zhu ZL, Xian HC, Wang HF, Chen BJ, Tang YJ, Tang YL and Liang XH (2021) Insight Into the Molecular Mechanism of Podophyllotoxin Derivatives as Anticancer Drugs. Front. Cell Dev. Biol. 9:709075
f. Kathleen Legarza and Li-Xi Yang. New Molecular Mechanisms of Action of Camptothecin-type Drugs. ANTICANCER RESEARCH 26 : 3301-3306 (2006) Review
2. Many references are missing in a manuscript e.g.chapter CURRENT CANCER THERAPY INVOLVING PHYTOCHEMICALS: Research has concentrated on finding ways to eliminate the consequences of the phytochemicals poor water solubility and its considerable hazardous side effects (reference??). Several attempts have been made to improve water solubility and tumor selectivity, leading to the synthesis of numerous analogs and pro drugs in this area (reference???).
3. Final conclusion is general and has already been repeated in numerous articles, especially for the phytochemicals which are described in this manuscript and already published papers: …….In conclusion, our analysis suggests that medications derived from natural products have the potential to be effective anticancer therapeutics and that additional research and testing are required to develop such alternatives to chemotherapy, those with fewer side effects but with equal or stronger efficacy.
Author Response
Reviewer comments 1
The manuscript is lacking in novelty. Manuscript text is a combination of articles that were already written. Role of phytochemicals, described in this manuscript (epipodophyllotoxin, camptothecin derivatives, taxane diterpenoids, vinca alkaloids, quercetin, curcumin, piperine, epigallocatechin gallate (EGCG), and gingerol) in cancer prevention are already described in detail in many articles, reviews.
Response 1
Thank you very much for your valuable comment. Although there are multiple reviews available on the anticancer and anti-inflammatory potential of the phytochemicals, but the need was constantly felt to provide the readers with an updated version of anticancer and anti-inflammatory activities of the most important phytochemicals on a single deck. We believe that our current manuscript will be a quick reference for the readers interested in anticancer as well as anti-inflammatory activities of the phytochemicals.
Reviewer comments 2
Many references are missing in a manuscript e.g., Chapter CURRENT CANCER THERAPY INVOLVING PHYTOCHEMICALS: Research has concentrated on finding ways to eliminate the consequences of the phytochemicals poor water solubility and its considerable hazardous side effects (reference??). Several attempts have been made to improve water solubility and tumor selectivity, leading to the synthesis of numerous analogs and pro drugs in this area (reference???).
Response 2
Thank you very much for your comment. The references have been added.
Reviewer comments 3
- Final conclusion is general and has already been repeated in numerous articles, especially for the phytochemicals which are described in this manuscript and already published papers: ……. In conclusion, our analysis suggests that medications derived from natural products have the potential to be effective anticancer therapeutics and that additional research and testing are required to develop such alternatives to chemotherapy, those with fewer side effects but with equal or stronger efficacy.
Response 3
This section has been updated, summarizing the content of the manuscript in two small paragraphs.

Reviewer 2 Report
The information presented is very relevant; however, it is recommended that some pertinent changes be made for a better understanding.
1.- In general, it is recommended to present the information with a numbering of the lines to allow making specific observations. It is also necessary to revise the wording of the words in vitro and in vivo throughout the manuscript, modifying them to italics.
Specifically, the following observations are made in the Taxanes section
In the section on taxanes, it is not clear if the information presented is in general for all taxanes (paclitaxel, docetaxel and cabazitaxel) because it only describes paclitaxel, include the negative effects of its use and establish the adequate concentrations.
3 .- Include information referring to high and low concentrations of this paragraph "Variable effects were observed depending on the amount of taxane used. Taxane therapeutic efficacy is proportional to the number of microtubules to which they are bound. "
CPT section
4.- It would be convenient to include the full name followed by the abbreviation ...... CPT
The description presented does not make clear the mechanism of action, although it presents a figure 2 that establishes the molecular basis, it does not present a clear description of the mechanism, it would be convenient to include it.
PTOX Section
6.- In the information presented for this compound, there are paragraphs in which too much information is described and it is not clear which is its bibliographic citation, specifically in the description of the structure and its possible mechanism of action, it is necessary to include it.
It would also be convenient to include a figure in which the molecular bases of action are described as it was described for CPT and Taxanes.
8 .- Why is separated in the section "Additional Anticancer Drugs Derived from Plants" to plants Euphorbia peplus (Euphorbiaceae) Cape bushwillow, or Combretum caffrum (Combretaceae), they are separated by mechanisms of action ????
9.- This section of "MECHANISMS OF CANCER CHEMOPREVENTION" the topics presented in this section could be the initial part of the review as described in the presented abstract. That is, to start with general information that describes an overview of the possible effects of phytochemicals and then indicate the compounds that are specifically described as those described in the section on "CURRENT CANCER THERAPY INVOLVING PHYTOCHEMICALS".
In the section of Inflammation vs Cancer, it would be convenient to include the molecular and signal transduction mechanism by which an inflammatory process can generate a cancer process, and then include information of the cases described.
It would be convenient to indicate why the authors only included colon and prostate cancer in this review.
12 Conclusion section
It is suggested to restructure the conclusion section because it is not presented in that context, it has an excess of information that could be considered at the beginning of the bibliographic review.
Author Response
Reviewer comment 1
1- In general, it is recommended to present the information with a numbering of the lines to allow making specific observations. It is also necessary to revise the wording of the words in vitro and in vivo throughout the manuscript, modifying them to italics.
Specifically, the following observations are made in the Taxanes section
In the section on taxanes, it is not clear if the information presented is in general for all taxanes (paclitaxel, docetaxel and cabazitaxel) because it only describes paclitaxel, include the negative effects of its use, and establish the adequate concentrations.
Response 1
Thank you very much for your comment. The wording of the words in vitro and in vivo throughout the manuscript is modified to italics. At the start we talk about taxanes but later we discuss only paclitaxel because of following reasons.
- To discuss other taxanes the length of the review paper would increase.
- Paclitaxel has been studied mostly in cancers and inflammation.
- Anyhow, to avoid confusion paclitaxel heading is added to the taxanes.
Regarding the concentrations and effects of taxanes, an extra paragraph highlighted in yellow color is introduced and a table 2. Please refer to taxane section.
Reviewer comment 2
CAMPTOTHECIN (CPT) section
4.- It would be convenient to include the full name followed by the abbreviation ...... CAMPTOTHECIN (CPT)
The description presented does not make clear the mechanism of action, although it presents a figure 2 that establishes the molecular basis, it does not present a clear description of the mechanism, it would be convenient to include it.
Response 2
Thank you very much for your comment. The full name followed by abbreviation is made clear. Also, a paragraph about the mechanism is added in the legends of the figure 3 highlighted in blue color plus further details on mechanism are also added to make it clearer highlighted in green, please refer to the section CAMPTOTHECIN (CPT) section.
Reviewer comment 3
PTOX Section
6.- In the information presented for this compound, there are paragraphs in which too much information is described, and it is not clear which is its bibliographic citation, specifically in the description of the structure and its possible mechanism of action, it is necessary to include it.
It would also be convenient to include a figure in which the molecular bases of action are described as it was described for CAMPTOTHECIN (CPT) and Taxanes.
Response 3
Thank you very much for your comment. The unnecessary paragraphs are removed. More relevant and accurate data is added. Please refer to the section “PTOX” highlighted in grey.
Reviewer comment 4
8.- Why is separated in the section "Additional Anticancer Drugs Derived from Plants" to plants Euphorbia peplus (Euphorbiaceae) Cape bushwillow, or Combretum caffrum (Combretaceae), they are separated by mechanisms of action????
Response 4
Thank you very much for your comment. Actually describing mechanisms for each and every compound would make it a very lengthy review paper, so we selected the most used and effected compounds in recent research and those with high efficacy.
Reviewer comment 5
9.- This section of "MECHANISMS OF CANCER CHEMOPREVENTION" the topics presented in this section could be the initial part of the review as described in the presented abstract. That is, to start with general information that describes an overview of the possible effects of phytochemicals and then indicate the compounds that are specifically described as those described in the section on "CURRENT CANCER THERAPY INVOLVING PHYTOCHEMICALS".
Response 5
Thank you very much for your comment. The data has been arranged accordingly. Please refer to the section "MECHANISMS OF CANCER CHEMOPREVENTION".
Reviewer comment 6
In the section of Inflammation vs Cancer, it would be convenient to include the molecular and signal transduction mechanism by which an inflammatory process can generate a cancer process, and then include information of the cases described.
Response 6
Thank you very much for your comment. The molecular and signal transduction mechanism by which an inflammatory process can generate a cancer process is added. Please refer to the section "inflammation vs cancer" highlighted in light grey.
Reviewer comment 7
It would be convenient to indicate why the authors only included colon and prostate cancer in this review.
Response 7
Thank you very much for your comment. We also discuss the skin cancer in above paragraphs but the only reason to limit our review to colorectal and prostrate is that mostly and more effectively phytochemicals are discussed in these two cancers.
Reviewer comment 8
12 Conclusion section
It is suggested to restructure the conclusion section because it is not presented in that context, it has an excess of information that could be considered at the beginning of the bibliographic review.
Response 8
This section has been updated, summarizing the content of the manuscript in two small paragraphs.

Round 2
Reviewer 1 Report
I agree with revised version of a manuscript.
There is an additional comment for the section- Conclusion:
Authors conclude: Phytochemicals could be utilized as supplements for the treatment and prevention of cancer.
If phytochemicals are used as supplements, attention must be paid to their bioavailability. For example, it is known that polyphenols are absorbed in different rates through the gastrointestinal tract. Certain chemical characteristics, for example, molecular weight, lipophilicity, stereochemistry, and the presence of a hydrogen-bonding group, affect the transport and permeability of the phytochemicals into the cytosol enterocytes from the gut lumen
e.g. quercetin as a potential component for health promotion is inadequately effective due to reduced bioavailability as a result of low absorption rate, low water solubility, and increased instability in alkaline and neutral media including various organs, such as the small intestine, colon and kidney.
Additionally there are often significant differences between the activities of the metabolic form of phytochemicals, e.g. phenols and their form in the nutraceutical/supplement matrix.
References:
1. Kozarski, M., Klaus, A., van Griensven, L., Jakovljevic, D., Todorovic, N., Wan-Mohtar, W.A.A.Q.I., Vunduk, J. (2023). Mushroom β-glucan and polyphenol formulations as natural immunity boosters and balancers: nature of the application, Food Science and Human Wellness, 12, 378-396.
2. M.B. Hussain, S. Hassan, M. Waheed, et al., Bioavailability and metabolic pathway of phenolic compounds, in M. Soto-Hernandez (Ed.), Plant physiological aspects of phenolic compounds, IntechOpen, London, 2019. https://doi.org 10.5772/intechopen.77494.
Author Response
I agree with revised version of a manuscript.
There is an additional comment for the section- Conclusion:
Authors conclude: Phytochemicals could be utilized as supplements for the treatment and prevention of cancer.
If phytochemicals are used as supplements, attention must be paid to their bioavailability. For example, it is known that polyphenols are absorbed in different rates through the gastrointestinal tract. Certain chemical characteristics, for example, molecular weight, lipophilicity, stereochemistry, and the presence of a hydrogen-bonding group, affect the transport and permeability of the phytochemicals into the cytosol enterocytes from the gut lumen
e.g. quercetin as a potential component for health promotion is inadequately effective due to reduced bioavailability as a result of low absorption rate, low water solubility, and increased instability in alkaline and neutral media including various organs, such as the small intestine, colon and kidney.
Additionally there are often significant differences between the activities of the metabolic form of phytochemicals, e.g. phenols and their form in the nutraceutical/supplement matrix.
References:
- Kozarski, M., Klaus, A., van Griensven, L., Jakovljevic, D., Todorovic, N., Wan-Mohtar, W.A.A.Q.I., Vunduk, J. (2023). Mushroom β-glucan and polyphenol formulations as natural immunity boosters and balancers: nature of the application, Food Science and Human Wellness, 12, 378-396.
- M.B. Hussain, S. Hassan, M. Waheed, et al., Bioavailability and metabolic pathway of phenolic compounds, in M. Soto-Hernandez (Ed.), Plant physiological aspects of phenolic compounds, IntechOpen, London, 2019. https://doi.org 10.5772/intechopen.77494.
Response: We highly appreciate the reviewer’s comment. The sentence containing the inappropriate word “supplements” has been removed from the conclusion section.

Reviewer 2 Report
The suggested corrections were made, however, I believe that one of the main ones was to list the lines to make specific observations, therefore, I am including the complete paragraphs where I suggest modifications be made.
1.- In this section it is necessary to include concentration measurements.
The overall concentration of taxanes inside cells greatly exceeds the concentration in the medium or plasma, as taxanes accumulate in cells and tumors [132, 133]. ∼20 μM of tubulin, or 4% of the total protein in HeLa cells, polymerizes in response to the addition of taxanes [134]. The relative importance of phenotypic effects on site occupancy was ranked by Pineda et al. Loss of MT dynamics was the most susceptible to perturbation, with detection starting at a site occupancy of ∼0.1. This was followed by micronucleation, which could be observed at a value of ∼0.2, and mitotic arrest, which required a value of 0.6-1.0. This provision was true for both cell lines and all four medications.
2.- The wording of this section needs to be modified because it appears to have been copied only from the original manuscript.
These high site occupancy values should be interpreted with caution because our model presumes homogeneous binding sites and extremely high drug concentrations can access other sites of lower affinity (e.g., at exposed ends).
3.- A revision of the whole manuscript is necessary where the words in vivo and in vitro are moridized in italic format.
4.- It is suggested that the bilbiographical quotations be included at the end of the sentence, not in the middle, for example:
Structural models demonstrate that camptothecin (CPT) binds to the Topo I-DNA binary complex in a non-covalent manner. Structure-activity correlations [177] provide insight into a possible mechanism by which camptothecin (CPT) and its derivatives decrease Topo I expression.
6.- It is not clear whether this section is part of the figure caption or should be moved to where
Figure 3 is indicated.
The ability of camptothecin to bind to double-stranded DNA is an important factor in its antitumor effect. During the replication process, these enzymes nick the double-stranded DNA, releasing a single-stranded copy. In addition, they significantly minimize supercoiling in the double helix structures of DNA. Camptothecin inhibits cancer growth by inhibiting the catalytic activity of topoisomerase-I after non-covalent binding, leading to elevated expression levels of p21, p53 and mTOR. In contrast, camptothecin stimulates ERK and NERF2, both of which are regulated by the drug. The cell death observed in Figure 3 was the result of these molecular processes, which triggered apoptosis [180, 181].
7.- In this section it is necessary to include the bibliographic citation of this information.
Other anticancer drugs derived from plants
a) It is a natural ester of the alkaloid cephalotaxin obtained from trees of the genus Cephalotaxus (Cephalotaxaceae). HHT limits protein synthesis by binding to the A-site cleavage of the large ribosomal subunit, which disrupts chain elongation.
b) The two naturally occurring chemicals are combretastatin A1 and A4. The orphan drug combretastatin A4 phosphate (CA4P) has been cleared by the FDA for the treatment of several types of thyroid and ovarian cancers.
Author Response
Reviewer comment 1
1.- In this section it is necessary to include concentration measurements.
The overall concentration of taxanes inside cells greatly exceeds the concentration in the medium or plasma, as taxanes accumulate in cells and tumors [132, 133]. ∼20 μM of tubulin, or 4% of the total protein in HeLa cells, polymerizes in response to the addition of taxanes [134]. The relative importance of phenotypic effects on site occupancy was ranked by Pineda et al. Loss of MT dynamics was the most susceptible to perturbation, with detection starting at a site occupancy of ∼0.1. This was followed by micronucleation, which could be observed at a value of ∼0.2, and mitotic arrest, which required a value of 0.6-1.0. This provision was true for both cell lines and all four medications.
Response 1
Thank you very much for your comment. The concentrations are added. Please see the section taxanes highlighted in yellow color.
Reviewer comment 2
2.- The wording of this section needs to be modified because it appears to have been copied only from the original manuscript.
These high site occupancy values should be interpreted with caution because our model presumes homogeneous binding sites and extremely high drug concentrations can access other sites of lower affinity (e.g., at exposed ends).
Response 2
Thank you very much for your comment. The wording of this section is modified. Please see the section taxanes highlighted in yellow color.
Reviewer comment 3
3.- A revision of the whole manuscript is necessary where the words in vivo and in vitro are moridized in italic format.
Response 3
Thank you very much for your comment. It has been revised thoroughly.
Reviewer comment 4
4.- It is suggested that the bilbiographical quotations be included at the end of the sentence, not in the middle, for example:
Structural models demonstrate that camptothecin (CPT) binds to the Topo I-DNA binary complex in a non-covalent manner. Structure-activity correlations [177] provide insight into a possible mechanism by which camptothecin (CPT) and its derivatives decrease Topo I expression.
Response 4
Thank you very much for your comment. The reference is moved to the end. Please see the section camptothecin (CPT) highlighted in green color.
Reviewer comment 5
6.- It is not clear whether this section is part of the figure caption or should be moved to where Figure 3 is indicated.
The ability of camptothecin to bind to double-stranded DNA is an important factor in its antitumor effect. During the replication process, these enzymes nick the double-stranded DNA, releasing a single-stranded copy. In addition, they significantly minimize supercoiling in the double helix structures of DNA. Camptothecin inhibits cancer growth by inhibiting the catalytic activity of topoisomerase-I after non-covalent binding, leading to elevated expression levels of p21, p53 and mTOR. In contrast, camptothecin stimulates ERK and NERF2, both of which are regulated by the drug. The cell death observed in Figure 3 was the result of these molecular processes, which triggered apoptosis [180, 181].
Response 5
Thank you very much for your comment. This is part of the figure caption.
Reviewer comment 6
7.- In this section it is necessary to include the bibliographic citation of this information.
Other anticancer drugs derived from plants
- a) It is a natural ester of the alkaloid cephalotaxin obtained from trees of the genus Cephalotaxus (Cephalotaxaceae). HHT limits protein synthesis by binding to the A-site cleavage of the large ribosomal subunit, which disrupts chain elongation.
- b) The two naturally occurring chemicals are combretastatin A1 and A4. The orphan drug combretastatin A4 phosphate (CA4P) has been cleared by the FDA for the treatment of several types of thyroid and ovarian cancers.
Response 6
Thank you very much for your comment. Proper references are added. Please see the section “Additional phytochemicals derived from plants” highlighted in light blue color.

Round 3
Reviewer 2 Report
Success in future work